# Recent Advances in Synergistic Antitumor Effects Exploited from the Inhibition of Ataxia Telangiectasia and RAD3-Related Protein Kinase (ATR)

**DOI:** 10.3390/molecules27082491

**Published:** 2022-04-12

**Authors:** Li-Wei Wang, Songwei Jiang, Ying-Hui Yuan, Jilong Duan, Nian-Dong Mao, Zi Hui, Renren Bai, Tian Xie, Xiang-Yang Ye

**Affiliations:** 1School of Pharmacy, Hangzhou Normal University, Hangzhou 311121, China; wangliwei78@163.com (L.-W.W.); 2019112012027@stu.hznu.edu.cn (S.J.); 2021112012261@stu.hznu.edu.cn (Y.-H.Y.); 2021112012301@stu.hznu.edu.cn (J.D.); 2020112012135@stu.hznu.edu.cn (N.-D.M.); huizi@hznu.edu.cn (Z.H.); 2Key Laboratory of Elemene Class Anti-Cancer Chinese Medicines, Engineering Laboratory of Development and Application of Traditional Chinese Medicines, Collaborative Innovation Center of Traditional Chinese Medicines of Zhejiang Province, Hangzhou Normal University, Hangzhou 311121, China

**Keywords:** ataxia telangiectasia and RAD3-related protein kinase (ATR), DNA damage response (DDR), inhibitor, synergistic effects, cancer therapy

## Abstract

As one of the key phosphatidylinositol 3-kinase-related kinases (PIKKs) family members, ataxia telangiectasia and RAD3-related protein kinase (ATR) is crucial in maintaining mammalian cell genomic integrity in DNA damage response (DDR) and repair pathways. Dysregulation of ATR has been found across different cancer types. In recent years, the inhibition of ATR has been proven to be effective in cancer therapy in preclinical and clinical studies. Importantly, tumor-specific alterations such as ATM loss and Cyclin E1 (CCNE1) amplification are more sensitive to ATR inhibition and are being exploited in synthetic lethality (SL) strategy. Besides SL, synergistic anticancer effects involving ATRi have been reported in an increasing number in recent years. This review focuses on the recent advances in different forms of synergistic antitumor effects, summarizes the pharmacological benefits and ongoing clinical trials behind the biological mechanism, and provides perspectives for future challenges and opportunities. The hope is to draw awareness to the community that targeting ATR should have great potential in developing effective anticancer medicines.

## 1. Introduction

From birth, each individual’s DNA is constantly exposed to damage from ultraviolet light, certain chemicals, toxins, and natural biochemical processes inside the cells. On the other hand, real DNA damage already starts in utero [1]. To cope with these DNA damages in the case of replication stress (RS), cells have a complex mechanism and machinery called DNA damage response (DDR) pathways, which are responsible for sensing (or detecting) the damaged DNA, engaging in cell-cycle adaptation, and performing damage repair processes. Such networks play a pivotal role to both normal cells and cancer cells to maintain their cell viability and avoid genomic instability [2,3]. Targeting DDR using DNA damaging agents is effective cancer therapy, but the relatively high toxicity typically limits its anticancer efficacy. Therefore, searching for compounds that could potentially exploit the synthetic lethality (SL) [4,5] of a DNA damaging mechanism has become a major ongoing focus in modern drug discovery.

There are six phosphatidylinositol 3-kinase-related kinases (PIKKs) family members, all of which are Serine/Threonine-specific protein kinases bearing a sequence similar to phosphatidylinositol-3 kinases (PI3Ks) [6]. Among them, ataxia-telangiectasia mutated kinase (ATM), ataxia-telangiectasia and RAD3-related kinase (ATR) [7], and DNA-dependent protein kinase catalytic subunit (DNA-PKcs) [8] are three important regulators centered in the DDR pathway. In 1996, scientists first discovered the 301.66 kDa protein ATR as the mammalian orthologue of mitosis entry checkpoint protein 1 (Mec1) in yeast [9]. This protein senses or detects DNA damage, and then stimulates the DNA damage checkpoint, further causing cell cycle arrest in eukaryotes. ATM and DNA-PKcs respond primarily to DNA double strand breaks (DSBs). In contrast, ATR reacts to single-stranded DNA (ssDNA) damage response, particularly the replication stress (RS).

The presence of persistent ssDNA damage is first detected by ATR kinase, which causes it to activate and further triggers the DNA repair process. Mechanistically, the replication protein A (RPA) coated on ssDNA will bind to the ATR-interacting protein (ATRIP), which will recruit ATR kinase to the site in close proximity to Rad9-Rad1-Hus1 (9-1-1). The ATRIP-ATR complex is capable to phosphorylate 9-1-1 subsequently. The phosphorylated 9-1-1 then binds to topoisomerase-binding protein 1 (TOPBP1), and a such binding will trigger the ATR-signaling cascade. It has been evidenced that ATR kinase is activated or upregulated in various cancer cells. The inhibition of ATR will interfere with the downstream signal transducer checkpoint kinase 1 (CHK1), leading to cell apoptosis (Figure 1). On the other hand, ATR is also capable of phosphorylating p53 tumor suppressor. Inhibition of ATR kinase could result in a p53-independent cell death due to premature chromatin condensation and mitotic entry. Numerous evidence from both in vitro and in vivo studies point to the fact that the inhibition of ATR kinase alone has great potential to treat cancer.

## 2. ATR Protein Structures

In 2017, Rao et al. [11] purified human ATR-ATRIP complex and solved its crystal structure at a resolution of 4.7 Å, using cryo-electron microscopy (EM). This suggests that the complex adopts a hollow “heart” shape, consisting of two ATR monomers in distinct conformations (Figure 2). Such conformational flexibility of ATR allows ATRIP to properly lock the N-termini of the two ATR monomers to favor complex formation and functional diversity. The two catalytic pockets of the kinase domains (KD, residues 2205–2615) symmetrically face outward and substrate access is not restricted by inhibitory elements. Clinical candidate berzosertib (VX-970, M6620, or VE-8222), an ATP-competitive ATR inhibitor developed by Vertex Co., binds with high affinity to the kinase domain of ATR in a docking experiment. The pyrain-2-amine group points towards the deep pocket of catalytic domain where the adenine (purine base) of ATP is supposed to be. The 5-membered ring of the isoxazole group is in a close proximity to the ribose of ATP [12]. It is worth mentioning that a 3.9 Å structure of the yeast Mec1-Ddc2 (a homolog of human ATR-ATRIP) complex was reported by Wang et al. [13]. In 2021, Burgers et al. [14] determined the structure of yeast Mec1(F2244L)-Ddc2 complex, a complex of single point mutant at residue 2244 of Mec1. This structure has much higher resolution of 2.8 Å than the one reported previously by Wang et al. [13]. Scientists were able to obtain near completely atomic models for Mec1-Ddc2 and uncover the molecular basis for low basal activity and the conformation changes required for activation. These results provide the structural basis for understanding ATR regulation and aid scientists in designing and discovering new ATR inhibitors for therapeutic usefulness. So far, the structure of a small molecule complexed with ATR kinase has not been obtained.

## 3. ATR Protein and Cancers

According to “The Human Protein Atlas” [15], ATR protein has low cancer specificity, which means that its expression is dysregulated in almost all cancer types. Thereby, ATR inhibitor has potential utility for the treatment of various cancers, either alone or in combination with other therapies. Many ATR inhibitors have been reported in publications or have been claimed in patent application in recent years [7]. Based on in vitro and in vivo studies, ATR inhibitors have the potential to treat the following cancers: blood cancers such as leukemia, myeloma and lymphoma; solid tumor such as breast cancer, gastric cancer, prostate cancer, pancreatic cancer, lung cancers (non-small cell lung cancer (NSCLC), small cell lung cancer (SCLC)), colorectal cancer, esophageal cancer, ovarian cancer, neuroendocrine carcinoma, hepatocellular carcinoma (HCC), biliary tract cancer, Ewing’s sarcoma, head and neck squamous cell cancer (HNSCC), soft tissue carcinoma, endometrial cancer, etc.

## 4. Synergistic Anticancer Effects Involving ATR Inhibition

Although fruitful modalities have been achieved to combat cancers, issues still exist. Firstly, specific oncogenes have proven undruggable by small molecules and antibodies [12]. Secondly, cancer cells have complex survival mechanisms, resulting in resistance development. Hartwell et al. first introduced the concept of SL when they identified novel vulnerabilities in cancer cells [16]. Now SL commonly refers to the synergistic effects in treating cancers when two genes are interfered or two therapeutic modalities are used concurrently in order to exploit a tumor-specific alteration [4,17,18,19,20]. Examples of such tumor-specific alterations include ATM loss [21,22], Cyclin E1 (CCNE1) amplification [23], and APOBEC3 expression [24]. These alterations render tumors more sensitive to ATR inhibition and could be exploited by SL strategy involving ATRi. Many other biomarkers are awaiting identification. Since ATR is centered in the DDR pathway, SL involving ATR inhibition might solve the problem with undruggable cancers and resistance. Recently Drew’s team briefly discussed some recent advances of SL involving ATR inhibition [25], but not extensively. Besides SL, an increasing number of therapeutic modalities have been reported to have synergistic anticancer effects when they are used in combination with an ATRi (Figure 3). This article provides an overview of up-to-date advances, hoping to provide the perspectives and challenges in maximizing ATRi-mediated synergistic anticancer effects in cancer treatment.

### 4.1. Radiotherapy (RT)

RT could damage DNA and stall cancer cells in the cell cycle. However, such treatment could be compensated by DDR mechanism and pathways to develop radioresistance. The search for a small molecule to sensitize RT (i.e., SL or synergistic anticancer effects when combining with RT) is urgently needed. In 2013, Bakkenist’s group showed that ATR activity in G_1_ phase cells facilitated the repair of IR-induced DNA damage [26]. This finding supports synergistic anticancer effects for combining RT with ATRi. Gollin’s team discovered that ATRi reversed radioresistance in oral squamous cell carcinoma (OSCC) [27]. Inhibition of cell cycle checkpoint ATR offers molecular differentiation between tumor and normal cells, thereby leading to synergistic anticancer effects. ATRi VE-821 (**1**, Figure 4) causes radiosensitization in several human tumor cells irradiated with high linear energy transfer (LET) radiation [28]. When ATRi ceralasertib and RT were used together in a NSCLC H460 mouse model, significant tumor growth delay was observed [29]. Similarly, ATRi berzosertib (**2**, Figure 4) enhanced the effect of radiation in NSCLC brain metastasis patient-derived xenografts (PDXs) models [30]. These results support the ongoing clinical trial of berzosertib in combination with whole brain irradiation in patients with NSCLC brain metastases. Tu et al. discovered that berzosertib significantly sensitized triple negative breast cancer (TNBC, high risk recurrence) PDXs to RT in vivo, while monotherapy of berzosertib showed no benefit [31]. The results reflect the SL between ATRi and RT. ATR inhibition was also found to sensitize the response of HNSCC to RT [32,33,34]. Ceralasertib (**3**, Figure 4) synergized with radioimmunotherapy by potentiating the tumor immune microenvironment in HCC [35]. It also potently radiosensitized p53-deficient NCI-H1299 cells at low concentrations [36]. All these results support the further clinical evaluation to exploit synergistic anticancer effects between ATRi and RT.

### 4.2. Platinum-Based Anticancer Drugs

Platinum-based anticancer drugs, including cisplatin, carboplatin, oxaliplatin, nedaplatin, and lobaplatin, are heavily used in chemotherapy regimens despite their well-known resistance and toxicity. Appropriate synergistic partners for them might overcome the resistance and could maximize anticancer efficacy at the same, reducing toxicity through a tailored dosing schedule. Genetic inhibition of ATR expression was found to enhance cisplatin sensitivity in human colorectal cancer (CRC) cells with inactivated p53 [37]. P53 mutations cause the loss of G1 checkpoint, therefore cells might be more vulnerable to inhibition of S/G2 DNA damage checkpoint signaling. Thereby, an ATRi in combination with cisplatin might be applied to p53-deficient tumors to minimize toxicity to normal tissues. In 2020, the first-in-class Phase I trial results of berzosertib as monotherapy or in combination with carboplatin in patients with advanced solid tumors were disclosed and preliminary anti-tumor responses were observed [38,39,40]. Other ATRi including berzosetib (**2**), elimusertib (**4**, Figure 4), ceralasertib (**3**), and gartisertib (**5**, Figure 4) are currently investigated in clinics exploiting synergistic anticancer effects with platinum-based drugs.

### 4.3. Topoisomerase I (TOP1) Inhibitors

TOP1 is an enzyme that controls and alters the topologic states of DNA during transcription; more specifically, the process of transient breaking and rejoining of a single strand of DNA. TOP1 is a validated drug target for cancer therapy. Camptothecin analogs irinotecan, belotecan, and topotecan are the effective FDA-approved anticancer agents from this class. Prommier et al. showed that VE-821 abrogated the S-phase replication elongation checkpoint and the replication origin-firing checkpoint induced by camptothecin [41]. Berzosertib enhanced the in vivo tumor response to irinotecan without additional toxicity. The results provide a rationale for combining TOP1 inhibitor and ATRi in clinical trials. Phase I results of berzosertib in combination with topotecan revealed that the combination was tolerable and particularly effective in platinum-refractory SCLC, which tends not to respond to topotecan alone [42]. The Phase II trial is currently ongoing (clinical trial ID: NCT04768296). Similarly, ATRi gartisertib significantly synergized with topotecan and irinotecan in patient-derived tumor organoids and xenograft models [43]. Recently, synergistic anticancer effects in chemotherapy-resistant ovarian cancer were achieved using the combination of ceralasertib and TOP1 inhibitor belotecan [44]. Additionally, the combination of TOP2 inhibitor etoposide and ceralasertib is currently in Phase II trial for extensive stage SCLC.

### 4.4. Nucleoside-Based Drugs

Pyrimidine nucleoside analog gemcitabine inhibits DNA synthesis and repair by incorporating a “faulty” base into DNA, leading to autophagy and apoptosis. Recently, Allan’s team reported that the combination of gemcitabine and berzosertib eradicated disseminated leukemia in an orthotopic mouse model, eliciting long-term survival and effective cure [45]. In a trial of patients with high grade serous ovarian cancer (HGSOC), the addition of berzosertib to gemcitabine only benefits the patients with replication stress (RS)-low tumors rather than those with (RS)-high tumors [46]. Konstantinopoulos et al. defined that the (RS)-high tumors as harboring at least one genomic RS alteration related to loss of retinoblastoma tumor suppressor (RB) pathway regulation and/or oncogene-induced replication stress. Such a RS biomarker should be used to design future clinical trials and patient stratification when the combination of an ATRi is used. Phase I results of berzosertib in combination with gemcitabine in patients with advanced solid tumors revealed preliminary efficacy signs [47]. There are at least five clinical trials ongoing exploiting synergistic anticancer effects between berzosertib and gemcitabine (clinical trial IDs: NCT04216316, NCT02567409, NCT04807816, NCT02627443, and NCT02595892). Moreover, Artios Pharma’s ATR0380 is currently in Phase I/II trials alone or in combination with gemcitabine in patients with advanced cancers (clinical trial ID: NCT04657068).

### 4.5. RNA Polymerase I (POL I) Inhibitors

RNA POL I is upregulated in acute myeloid leukemia (AML) cells. POL I inhibitor could be a potential AML therapy. CX-5461 is a potent POL I transcription inhibitor and stabilizer of the DNA G-quadruplex structure. It induces cell apoptosis partially through intrinsic apoptotic pathway and is independent of TP53 status. Taub et al. reported that the combination of POL I inhibitor clinical candidate CX-5461 and ceralasertib synergistically inhibited the cell proliferation in AML cell lines CTS and U937 [48]. The addition of ceralasertib causes abolishment of the G2/M cell cycle checkpoint arrest, leading to better and synergistic anticancer effects. The results provide proof-of-concept for combining these two clinical candidates in future trials. It should be noted that Pritchard et al. recently reannotated CX-5461 as a TOP2 inhibitor [49].

### 4.6. PARP Inhibitors

Poly (ADP-ribose) polymerase (PARP) is a family of proteins mediating a number of cellular processes such as DNA repair, genomic stability, and programmed cell death. Up to now, there are four PARP inhibitors approved by the US FDA: olaparib, rucaparib, niraparib, talazoparib. Within the synergistic anticancer effects field involving ATR inhibition, PARP is probably the target that has received the most intensive studies.

#### 4.6.1. Ceralasertib

In 2020, Simpkins et al. developed PARPi and platinum-resistant in vitro and in vivo models from germline BRCA1/2 mutant patient tumors. Using these models, they further validated that the combination of PARPi and ATRi synergistically decreased cell viability in vitro and suppressed tumor growth in vivo. The combination of the two inhibitors also significantly prolonged the overall survival of the animals [50]. The synergistic anticancer effects provided mechanistic support and led to the design of the CAPRI trial (clinical trials ID: NCT03462342) for evaluating ceralasertib and olaparib in recurrent ovarian cancer patients [51]. The synergistic anticancer effects between ceralasertib and olaparib in ATM loss xenograft and PDX mouse models was also reported by Young’s group [52]. The preliminary results of ceralasertib–olaparib arm of a clinical trial called OLAPCO (clinical trial ID: NCT02576444) revealed that the overall response rate was 8.3%, and the clinical benefit rate (CBR) was 62.5% [53]. Yale scientists reported that the isocitrate dehydrogenase-1/2 (IDH-1/2) mutations confer a homologous recombination deficiency (HRD) phenotype and, hence, are sensitive to both PARPi and ATRi. The combination of ceralasertib and olaparib effectively inhibits the growth ofmutant cancer cells in vitro [54]. The combination caused significant tumor shrinkage in the mice model compared to monotherapy. The results support ongoing clinical trials in patients with IDH-1/2 mutant solid tumors (clinical trial ID: NCT03878095). Up to now, there are at least 17 clinical trials listed in clinicaltrials.gov website for ceralasertib–olaparib combination.

#### 4.6.2. Elimusertib

Elimusertib is a clinical stage ATRi developed by Bay Company. The first-in-human Phase I dose-escalation trial showed elimusertib was well tolerated with clinical evidence of anti-tumor activity in patients with advanced cancers with ATM aberrations [55]. Co-administration of elimusertib (50 mg/kg) and olaparib (30 mg/kg) in an ATM loss patient-derived prostate cancer model resulted in better tumor growth inhibition and longer survival of animals [56]. Wengner et al. also reported that the combination of elimusertib and olaparib led to synergistic anti-tumor activity in vivo [57]. The combination of elimusertib and niraparib is currently in Phase I clinical trial for ovarian cancer and other advanced solid tumors (clinical trial ID NCT04267939).

#### 4.6.3. RP-3500

RP-3500 is a clinical compound developed by Repare Therapeutics. The combination of RP-3500 with a reduced dose of olaparib or niraparib was found to maximize tumor growth inhibition, while minimizing the impact on red blood cell depletion [58]. These results strongly support ongoing clinical trials of RP-3500 (clinical trial ID: NCT04972110 and NCT04497116).

#### 4.6.4. M1774

M1774 is a clinical stage ATRi developed by Merck KGaA. The Phase I trial is designed to examine M1774 alone or in combination with niraparib in patients with metastatic or locally advanced unresectable solid tumors (clinical trial ID: NCT04170153) [59]. The structure of M1774 is not disclosed publicly.

### 4.7. PI3K Inhibitors

Phosphoinositide 3-kinases (or phosphatidylinositol 3-kinases, PI3Ks) are enzymes capable of phosphorylating the 3-position hydroxyl group of the inositol ring of phosphatidylinositol. They are anticancer drug targets due to their function in modulating cell growth, proliferation, differentiation, motility, survival and intracellular trafficking. Inhibitors such as idelallisib, copanlisib, duvelisib, alpelisib, and umbralisib have been approved for clinical uses. In 2021, Guo et al. reported that the combination of PI3K inhibitor NVP-BEZ235 with berzosertib increased cytotoxicity in human esophageal cancer (ESCC) cells bearing Neuritin 1 (NRN1) repressed by promoter region methylation [60]. The combination of NVP-BEZ235 and berzosertib significantly reduced tumor growth in an ESCC cell xenografts mice model. The methylation of the promoter of the NRN1 gene causes epigenetic silencing, and it is a novel marker for the synergistic anticancer effects observed between PI3Ks inhibitor and ATRi. Furthermore, the combination of elimusertib and copanlisib is being evaluated in Phase I trial in patients with advanced solid tumors (clinical trial ID: NCT05010096).

### 4.8. AXL Receptor Tyrosine Kinase Inhibitors

AXL receptor tyrosine kinase (AXL) is a receptor tyrosine kinase encoded with AXL gene. It transduces signals from the extracellular matrix into the cytoplasm by binding growth factors such as the vitamin K-dependent protein growth-arrest-specific gene 6 (GAS6). Gliteritinib is a dual AXL-FLT3 inhibitor approved by the US FDA for the treatment of AML. In 2020, Byers et al. reported that the combination of AXL inhibitor and ATRi significantly decreased cell proliferation of NSCLC and large cell neuroendocrine carcinoma (LCNEC) cell lines [61]. Particularly, NSCLC cell lines with low levels of schlafen family member 11 (SLFN11) were more sensitive to AXL/ATR co-targeting treatment, reflecting SL between the two targets.

### 4.9. BAF Inhibitors

BAF complex, or SWItch/Sucrose Non-Fermentable (SWI/SNF) complex, is one of the ATP-dependent chromatin remodeling complexes found in eukaryotes. About 20% of human cancers have BAF mutation. Crabtree et al. reported that BAF inhibitor BRD-K98645985 (EC50 2.37 μM) synergized with VE-821 in killing cancer cells [62]. The combination of the two drugs might force cells into mitotic catastrophe and subsequent cellular arrest, leading to synergistic anticancer effects.

### 4.10. CHK1 Inhibitors

Check point kinases are often aberrantly regulated in cancers, including AML. Checkpoint kinase 1 (CHK1) is the major downstream effector of ATR. The strategy utilizing short interfering RNA screen directed against cell cycle and DNA repair genes was developed by Tibes’s team in 2014 to identify SL between WEE1 inhibitor (WEE1i) MK1775 and VE-821 [63]. Since then, several papers related to SL of ATRi and CHK1 inhibitors have been published and were included in the review by Zhang’s team [64]. In 2016, Helleday et al. reported that the combination of ATRi with CHK1 inhibitor resulted in synergistic anticancer effects [65]. In 2019, Carrassa et al. reported that the combination of ceralasertib with AZD-7762 exerted a strong synergistic cytotoxic effect in two lymphoma subtypes, regardless of p53, MYC, and ATM mutational status [66].

### 4.11. XPO1 Inhibitors

Exportin 1 (XPO1), i.e., chromosomal region maintenance 1 (CRM1), is a eukaryotic protein that mediates the nuclear export of various proteins and RNAs. It is involved in various viral infections and some cancers. Selinexor, a drug specifically targeting XPO1, was approved by the US FDA for the treatment of multiple myeloma. In 2021, Carugo’s team reported that XPO1 inhibition served as drivers of DNA damage-induced lethality in TP53-mutant CRC [67]. Administration of XPO1 inhibitor KPT-8602 followed by ceralasertib resulted in dramatic anti-tumor effects and prolonged survival in TP53-mutant models of CRC.

### 4.12. WEE1 Inhibitor

The 96 kDa WEE1 is a member of the Ser/Thr family of protein kinases and a key regulator of cell cycle progression, specifically G2/M checkpoint transition. WEE1 inhibition could result in high CDK1 activity and cell progression through the G2/M checkpoint without adequately repairing DNA damage, thus generating mitotic catastrophe and cell death. However, monotherapy for WEE1i in treating cancers encounters certain limitations. Therefore, the SL between WEE1i and ATRi could be the future direction for anticancer drug discovery. As early as 2014, the SL between WEE1i and ATRi was reported in AML [63]. Such SL was also reported by Gordon et al. recently in Ewing sarcoma cells [68]. Wang’s team reported that the combination of WEE1i AZD1775 and ceralasertib promoted the accumulation of cytosolic double-strand DNA, which subsequently activated the stimulator of the interferon gene (STING) pathway and induced the production of type I interferon and CD8+ T cells, thereby inducing anti-tumor immunity [69]. Additionally, the team observed that blocking programmed death-ligand 1 (PD-L1) enhanced the effects of such a combination, further validating the WEE1i-ATRi-PD-L1 antibody triple combination effectiveness for cancer treatment. SL effects between AZD1775 and ceralasertib were also observed in U2OS osteosarcoma cells and in four lung cancer cell lines [70]. Simpkins et al. identified oncogene Cyclin E1 (CCNE1) as the important biomarker for predicting responsiveness to low-dose WEE1i-ATRi combination in aggressive subsets of ovarian and endometrial cancers [71]. In ovarian and endometrial cancer models, animal group with CCNE1 amplification had much longer survival rate than the CCNE1 low group. AZD1775 combined with ceralasertib exerted more potent anti-tumor effects against biliary tract cancer than either drug alone in both in vitro and in vivo studies. Furthermore, the SL effects between berzosertib and AZD1775 were also reported in AML cell lines [72].

### 4.13. ALDH Inhibitors

Aldehyde dehydrogenase (ALDH) enzymes are found to over-express in cancer cells and are associated with certain drug resistance. ALDH enzymes protect cells by metabolizing toxic aldehydes which can induce DNA double stand breaks (DSB). Therefore, ALDH inhibitor (ALDHi) could demonstrate potential utility in cancer therapy. Recently Buckanovich et al. reported that the combination of ALDHi 673A and ceralasertib resulted in synergistic killing of ovarian cancer cell lines [73].

### 4.14. HSP90 Inhibitors

Heat shock proteins (HSP) are families of proteins that are produced by cells in response to stressful conditions. There are a number of HSP inhibitors in clinical stages for potential cancer treatment. The SL effects between HSP90 inhibitor AUY922 and VE-821 were reported recently in Ewing’s sarcoma (ES) cells [74]. These results might provide support for further evaluation of the HSP90 inhibitor-ATRi combination in clinics.

### 4.15. HDACs Inhibitors

Histone deacetylases (HDACs) are a class of enzymes that remove acetyl groups from an ε-N-acetyl lysine amino acid on a histone, allowing the histones to wrap the DNA more tightly [75]. Five HDACs inhibitor have been approved for clinical uses. In a patent application published in 2019, HDACs inhibitor belinostat was found to synergize with ATR inhibitor ceralasertib in MDA-MB231, U973, and MCF-10A cells [76]. Such SL effects could potentially expand the indications of HDACs inhibitors to broader diseases such as solid tumors.

### 4.16. BET Inhibitors

Bromodomain and extraterminal domain (BET) family proteins (BRD2, BRD3, BRD4, and BRDT) are a family of epigenetic readers that recognize acetylated-lysine residues on histones and non-histone chromatin factors. BRD4 is the best studied subtype that has been implicated in various human cancers. In 2016, Nilsson’s team reported that the ATRi and BET inhibitor (BETi) combination caused robust transcriptional changes in genes involved in cell death, senescence-associated secretory pathway, NF-κB signaling, and endoplasmic reticulum (ER) stress [77] Subsequently, they reported that the SL antitumor effects between ATRi and BETi against melanoma, both in vitro and in vivo [78]. Chen et al. uncovered that BRD4 regulated the function of CDC6 and played an indispensable part in DNA replication checkpoint signaling [79]. Inhibition of BRD4 could lead to a reduction in CHK1 phosphorylation. Since the SL effects existed between CHK1i and ATRi, it is understandable that BRD4 inhibition synergized with ceralasertib in killing a number of cancer cell lines. Strong synergistic anticancer effects were observed in ovarian cells patient-derived xenograft models. Bertoni et al. reported that the combination of BETi birabresib and ceralasertib was the most active among other combinations in several types of mantle cell lymphoma [80]. The synergistic anticancer effects between BRD4 inhibitor JQ1 and ceralasertib were also reported in leukemia cells [81].

### 4.17. Aurora Kinase A Inhibitors

Aurora kinase A (or Aurora A, serine/threonine protein kinase 6) is activated by one or more phosphorylations. Its activity peaks during the G2 phase to M phase transition in the cell cycle. Dysregulation of Aurora A has been associated with a high occurrence of cancer. Eilers reported that the combination of Aurora A inhibitor alisertib and ceralasertib induced rampant tumor-specific apoptosis and tumor regression in mouse models of neuroblastoma, leading to permanent eradication in a subset of mice [82].

### 4.18. BUB1 Inhibitor

Budding uninhibited by benzimidazoles 1 (BUB1) is a mitotic checkpoint serine/threonine protein kinase that plays a key role in establishing the mitotic spindle checkpoint and chromosome congression. In 2018, the synergistic anticancer effects were first reported by Bayer Pharma in various tumor cell lines [83]. In the following year, Bayer scientists reported that BUB1 inhibitor BAY 1816032 sensitized tumor cells toward ATRi AZ20 [84].

### 4.19. GLUT1 Inhibitors

In mammalian cells, the uniporter protein glucose transporter 1 (GLUT1) facilitates glucose transport across the plasma membranes in a single direction. Erber’s team identified a novel synergistic interaction between GLUT1-mediated glucose transport and the cell cycle checkpoint kinase ATR [85]. The combination of GLUT1 inhibitor WZB117 and berzosertib robustly induced apoptosis, particularly in RAS-mutant cancer cells. These results were translated to robust tumor suppression in an autochthonous mouse model of KRASG12D-driven soft tissue sarcoma in vivo.

### 4.20. CDC7 Inhibitors

Cell division cycle 7-related protein kinase (CDC7) is an enzyme involved in the regulation of the cell cycle at the point of chromosomal DNA replication. Santocanale et al. reported that CDC7 and ATR co-inhibition led to premature and highly defective mitosis [86]. Through integrated bioinformatics analyses and a non-biased CRISPR loss of function genetic screen, Bernards et al. discovered that CDC7 inhibitor XL413 sensitized HCC cells towards ceralasertib treatment in vitro and in vivo [87]. The SL between ATR inhibition and CDC7 inhibition probably derives from abnormalities in mitosis, inducing mitotic catastrophe.

### 4.21. ALK Inhibitors

Anaplastic lymphoma kinase (ALK), also known as ALK tyrosine kinase receptor or CD246 (cluster of differentiation 246), is an enzyme that plays a pivotal role in cellular communication and in the normal development and function of the nervous system. Several ALK inhibitor (ALKi) drugs have been approved by the US FDA. The synergistic anticancer effects between ATRi and ALKi were reported by Palmer’s team in neuroblastoma (NB) models. The combination of elimusertib (3 day on/4 day off) and lorlatinib in 14-day treatment resulted in complete tumor growth suppression in two different mouse models with no detectable side effects [88]. These results support conducting combination studies in NB patients, particularly in high-risk groups with oncogene-induced replication stress.

## 5. Conclusions and Perspectives

Many therapeutic modalities have been reported to have synergistic anticancer effects with ATRi. Among them, the combination with PARP inhibitors has received most extensive studies so far. The results discussed here could provide the drug discovery community with more options and a clearer direction for developing ATRi anticancer drugs, as well as optimal combination strategies involving ATRi. In fact, many ongoing clinical trials have exploited the synergistic anticancer effects of ATRi and other therapeutic modalities. The area should attract greater attention in the coming future in drug discovery.

Despite the great progress that has been made in the field, challenges are still ahead of us in terms of how to maximize synergistic anticancer effects. Identification of the biomarkers or mutations that render cancer cells more sensitive to ATR inhibition or the combination is the key. First of all, the mechanism of SL between ATR inhibition and other therapeutic modalities requires in-depth studies, although the literature has disclosed some. Secondly, the biomarkers for selecting better synergistic anticancer effects should be further investigated and understood. This will allow us to better stratify certain patient populations to maximize clinical benefits. Thirdly, the potential increase in toxicity should be watched. With the synergistic anticancer effects in action, toxicity might increase. In fact, some animal studies required specially tailored dosing schedules such as certain days on and certain days off.

Overall, research exploiting synergistic anticancer effects between ATRi and other therapeutic modalities has drawn a substantial increase in attention in today’s drug discovery in recent years. Identification of the specific biomarkers such as ATM mutations, CCNE1 expression or APOBEC3 expression in tumors will allow the confirmation of tumor types better sensitive to ATR inhibition. With eight ATRi clinical candidates (berzosertib, ceralasertib, elimusertib, gartisertib, M1774, ATRN-119, RP-3500, and ART0380) being investigated currently, the synergistic anticancer effects listed here should provide better combination options for the clinician and drug discovery community to maximize the usefulness of an ATRi.

## Figures and Tables

**Figure 1 molecules-27-02491-f001:**
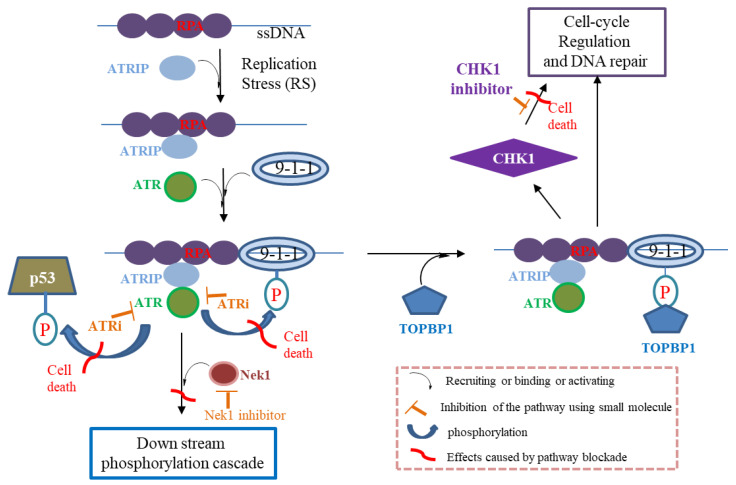
ATR inhibitor prevents the phosphorylation of 9-1-1 and p53 tumor suppressor, causes downstream signaling including cell apoptosis and cell cycle arrest. Abbreviations: ATR, ataxia-telangiectasia and RAD3-related kinase; ATRi, ATR inhibitor; ATRIP, ATR-interacting protein; CHK1, checkpoint kinase 1; ssDNA, single-stranded DNA; Nek1, NIMA (never in mitosis A)-related kinase 1 [10]; P, phosphorylation; RPA, replication protein A; RS, replication stress; TOPBP1, topoisomerase-binding protein 1; 9-1-1, Rad9-Rad1-Hus1. Adapted with permission from Ref. [7]. 2022, Expert Opinion on Therapeutic Patents.

**Figure 2 molecules-27-02491-f002:**
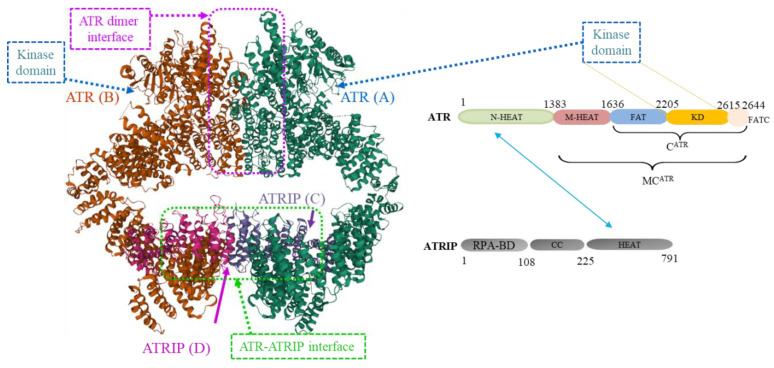
Crystal structure of dimerized ATR-ATRIP complex (PDB code: 5YZ0) [11]. ATR (A) represents one of the ATR protein of the dimer; ATR (B) represents the other ATR protein of the dimer; ATRIP (C) represents the ATRIP protein interacting with ATR (A); ATRIP (D) represents the ATRIP protein interacting with ATRIP (D). Abbreviations: RPA-BD, Replication Protein A (RPA)-binding domain; CC, coiled coil; N-HEAT, N-terminal HEAT repeats; M-HEAT, middle HEAT repeats; FAT, refers to “FRAP, ataxia-telangiectasia mutated kinase (ATM), and TRRAP domain”; KD, kinase domain; FATC refers to “FRAP-TM-TRRAP-C-terminal” domain.

**Figure 3 molecules-27-02491-f003:**
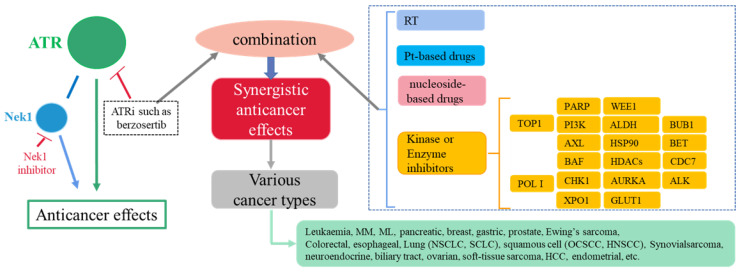
Synergistic anticancer effects between ATRi and other therapeutic modalities. Abbreviations: ALDH, aldehyde dehydrogenase; ALK, anaplastic lymphoma kinase; ATRi, ATR inhibitor; AURKA: Aurora-A kinase; AXL, AXL receptor tyrosine kinase; BAF, SWItch/Sucrose Non-Fermentable (SWI/SNF) complex; BET, Bromodomain and extra-terminal domain; BUB1, Budding uninhibited by benzimidazoles 1; CDC7, Cell division cycle 7-related protein kinase; CHK1, Checkpoint kinase 1; GLUT1, glucose transporter 1; HDACs, histone deacetylases; HSP90, heat shock protein 90; ML, malignant lymphoma; MM, multiple myeloma; HCC, hepatocellular carcinoma; HNSCC, head and neck squamous cell carcinoma; Nek1, NIMA (never in mitosis A)-related kinase 1; NSCLC, non-small cell lung cancer; OCSCC, oral cavity squamous cell carcinoma; PARP, Poly (ADP-ribose) polymerase; PI3K, Phosphoinositide 3-kinases; POL I, RNA Polymerase I, Pt-based: Platinum-based; RT, radiotherapy; SCLC, small cell lung cancer; TOP1, Topoisomerase I; WEE1, Wee1 protein kinase; XPO1, Exportin 1.

**Figure 4 molecules-27-02491-f004:**
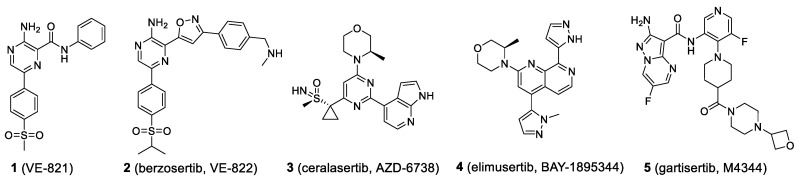
Structures of known ATR inhibitors: ATR inhibitor lead compound VE-821 and four clinical candidates developed by various pharmaceutical companies.

## Data Availability

Not applicable.

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
