# Peer review of "Recent Advances in Synergistic Antitumor Effects Exploited from the Inhibition of Ataxia Telangiectasia and RAD3-Related Protein Kinase (ATR)"

_molecules, 2022, doi:10.3390/molecules27082491_

Round 1
Reviewer 1 Report
This is a nicely written, focused and short review on “Recent advances of synergistic antitumor effects exploited from the inhibition of ataxia telangiectasia and RAD3-related protein kinase (ATR)” by Wang et al. I would suggest publication following minor revisions.
Figure 1:
RPA is missing in the figure (clarify the blue dots ATRIP is binding to)
Figure legend needs to be extended to clarify better what happens also without addition of ATRi.
Figure 2:
Abbreviations for FAT, FRAP, ATM, TRRAP are not explained
Figure 3:
Pt-based, AXL, BAF, WEE1, Aurora-A, BET are missing in the figure legend.
Also arrange the order of abbreviations like in they appear in the figure (BUB1 comes after GLUT1)
Generally, the figure would be more informative, if also the downstream target signaling molecules are integrated into graphical pathway flows.
Figure 4:
Why is especially Vertex company important? Why only this company is mentioned in the figure legend.
Also referring to conclusion “with eight ATRi clinical candidates” – why are not these 8 shown (only 5)?
Text:
Citation is sometimes not complete. For example, chapter 5 without any citation is meaningless! Also the stage of clinical development should be given, or, if already mentioned earlier, the whole chapter should be removed.
In general, explain abbreviations in the text when they appear for the first time (not just only in the figure legend). E.g. NSCLC, SCLC line 110, TNBC line 164, RB line 217, AXL line 304, SLFN11 line 310, BAF line 314, ER line 397
Line 187 (Figure 2 is missing for berzosetib, ceralasertib)
Line 217 typo RB? If RB, explain what is meant!
Line 243 US FDA – later you only use FDA, up to my knowledge FDA is anyway in US
Line 243 clarify if three or four PARPi are approved
Line 481 which eight ATRi?
Author Response
Dear Reviewer,
Thank you so much for your time and effort. We have revised the manuscript based on your suggestion. Please see the letter to you in the attachment. Thank you once again!

Reviewer 2 Report
This is an interesting and potentially very valuable review contribution by Wang and coworkers, on the role of ATR in DDR and its exploration as a target in clinical cancer settings. However, there are some concerns on the figures, text and work cited and discussed that should be addressed apropriatelty before publication can be fully recommended.
Major points:
- Figures are nice in general but some of them have really sucint legends, making their understanding rather dificult. Example : Figure 1: do the different shapes and colours of the arrows used have distinct meanings? What are they: protein protein interaction, activation, inhibition, phosphorylation, protein degradation . All should be really explained in detail in the legend. Also , since in the text there is not much discussion of this figure, the legend should walk the reader through the diagram in the legend.
- Fig 1 and 3 and associated text: the authors left out an important finding on the upstream activation of ATR by NEK1 kinase. The relevant work should be cited and Nek1 can be integrated in the figures 1 and 3: Liu et al., 2013, PNAS 110(6): 2175-80, "Nek1 associates with ATR/ATRIP and primes ATR for damage response...."
- English language use is sometimes cumbersome and should be revised by expert: examples: line 32 introduction : "Since born, each ...." better: "Since birth, each..." . On the other hand , really DNA damage already starts in utero ....! other example: line 80 , "fit nicely", not very scientific expression, "Vertex bind with high affinity to kinase domain?"line 88: "it allows scientists to .... ", strange expression in this context
Author Response
Dear Reviewer,
Thank you so much for your time. We have incorporated your valuable suggestion in the revised manuscript.

Round 2
Reviewer 2 Report
The authors responded well all raised points and incorporated all changes in a sucessful fashion in the text. Figures and legends also improved substantially.
Therefore, I now recommend acceptance of this interesting and valuable review.
This manuscript is a resubmission of an earlier submission. The following is a list of the peer review reports and author responses from that submission.